# The Multisystem Impact of Long COVID: A Comprehensive Review

**DOI:** 10.3390/diagnostics14030244

**Published:** 2024-01-24

**Authors:** Nicoleta Negrut, Georgios Menegas, Sofia Kampioti, Maria Bourelou, Francesca Kopanyi, Faiso Dahir Hassan, Anamaria Asowed, Fatima Zohra Taleouine, Anca Ferician, Paula Marian

**Affiliations:** 1Department of Psycho-Neuroscience and Recovery, Faculty of Medicine and Pharmacy, University of Oradea, 410073 Oradea, Romania; 2Department of Orthopaedics, Achillopouleio General Hospital of Volos, Polymeri 134, 38222 Volos, Greece; geomenegas@gmail.com; 3Faculty of Medicine and Pharmacy, University of Oradea, 410073 Oradea, Romaniambourelou@gmail.com (M.B.); faaydx88@gmail.com (F.D.H.);; 4University College London Hospitals NHS Foundation Trust, 250 Euston Road, London NW1 2PG, UK; ftaleouine@hotmail.com; 5Department of Medical Disciplines, Faculty of Medicine and Pharmacy, University of Oradea, 410073 Oradea, Romania; anca.moza@yahoo.com (A.F.);

**Keywords:** Long COVID, COVID-19, SARS-CoV-2, hypercoagulability, dysbiosis

## Abstract

(1) Background: COVID-19 was responsible for the latest pandemic, shaking and reshaping healthcare systems worldwide. Its late clinical manifestations make it linger in medical memory as a debilitating illness over extended periods. (2) Methods: the recent literature was systematically analyzed to categorize and examine the symptomatology and pathophysiology of Long COVID across various bodily systems, including pulmonary, cardiovascular, gastrointestinal, neuropsychiatric, dermatological, renal, hematological, and endocrinological aspects. (3) Results: The review outlines the diverse clinical manifestations of Long COVID across multiple systems, emphasizing its complexity and challenges in diagnosis and treatment. Factors such as pre-existing conditions, initial COVID-19 severity, vaccination status, gender, and age were identified as influential in the manifestation and persistence of Long COVID symptoms. This condition is highlighted as a debilitating disease capable of enduring over an extended period and presenting new symptoms over time. (4) Conclusions: Long COVID emerges as a condition with intricate multi-systemic involvement, complicating its diagnosis and treatment. The findings underscore the necessity for a nuanced understanding of its diverse manifestations to effectively manage and address the evolving nature of this condition over time.

## 1. Introduction

Severe acute respiratory syndrome coronavirus 2 (SARS-CoV-2) causes the disease we now call COVID-19. In early December 2019, the first cases of COVID-19 started to appear, igniting the beginning of a pandemic that has, by the time this article is written, taken 6.9 million lives worldwide. The report of the fifteenth meeting of the International Health Regulations Emergency Committee, which was held on the 4 May 2023, concluded with the formal end of the public health emergency regarding the COVID-19 pandemic. With a decrease in deaths and hospitalizations due to acute COVID-19, interest has now slowly shifted towards the long-term complications seen in both prolonged and recovered cases. Thus, the term Long COVID has emerged, encompassing multiple symptoms that manifest after a SARS-CoV-2 infection. Long COVID places additional pressure on organs that are already strained by the acute form of the disease, resulting in multi-systemic long-term symptoms. Long COVID, according to the World Health Organization (WHO), represents any clinical manifestation present in the first three months after acute COVID-19, minimally evolving over two months, without another identifiable cause [1].

The symptomatology comprises a wide variety of symptoms, some more frequently encountered, such as fatigue, exertional dyspnea, insomnia, malaise, cognitive impairment, myalgia, cough, anosmia, arthralgia, chest pain, fever, tachycardia, and palpitations [2,3,4,5,6,7,8,9,10,11,12,13,14,15,16]. Accurately establishing the epidemiological risk factors is highly important, as they are extremely valuable both for diagnostic and research purposes. Most of the articles that were reviewed link a high incidence of manifestation or persistence of Long COVID symptoms in females [2,3,4,5,6,7,8,9,10,11,12,14,15], older patients [4,7,15], patients with previous pulmonary diseases [4,7,9,10,15], diabetic patients [15,16], obese patients [3,4,15], patients with previous cardiovascular pathology [15,16], patients with increased SARS-CoV-2 viral load [8], smokers [3,4], patients with depression [4,5,6,7,8,9,10,11], and patients that experienced severe COVID-19 [7,8,9,13,14].

This research aims to comprehensively detail and provide an understanding of the clinical manifestations of Long COVID, a newly emerged medical entity recognized in the aftermath of the pandemic. The research focuses on the involvement of specific systems or organs, with an emphasis on identifying and characterizing the unique features of these manifestations.

## 2. Materials and Methods

References were found in relevant databases such as the National Center for Biotechnology Information (PubMed), the Centers for Disease Control and Prevention, the Scientific Electronic Library Online, Scopus, Web of Science, and the Cochrane Database. For this review, utilizing searches within the publication interval of 1 January 2021–1 December 2023, along with linking the term “Long Covid” with one of the following: “cardiovascular diseases or renal diseases or respiratory diseases or gastrointestinal diseases or neuro-psychiatric diseases or endocrine diseases or skin diseases or pathology or dysbiosis or chronic fatigue syndrome or myalgic encephalomyelitis”. The obtained articles underwent examination. Their selection and rejection were carried out according to the flowchart represented in Figure 1.

## 3. The Multisystem Impact of Long COVID

### 3.1. Pulmonary Involvement in Long COVID

The main impact of SARS-CoV-2 infection is on the respiratory system; the entrance of the virus into the epithelial lining of the lungs is facilitated by pneumonocytes that express the angiotensin-converting enzyme 2 (ACE2) receptors [17]. Upon infection by SARS-CoV-2, lung epithelial cells, which serve as the primary origin of inflammatory cytokines, engage in interactions with immune cells that have been recruited to the site of infection. This interaction plays a significant part in the development of inflammatory lung damage and subsequent respiratory failure [18]. The persistence and unregulated activation of cytokine storms can result in the impairment of the epithelial barrier. In the presence of such circumstances, the progression of lung fibrosis occurs due to the impairment of lung epithelial regeneration [19].

The injury inflicted by SARS-CoV-2 on a range of lung epithelial cells is attributed to the ACE2 receptors or transmembrane serine protease 2 (TMPRSS2) expression. ACE2 has been observed to be expressed in various types of cells in the airway epithelium, such as basal cells, ciliated cells, mucous cells, club cells, and intermediate cells [20]. According to a single-cell ribonucleic acid (RNA) sequencing analysis, it was observed that the average amount of ACE2 was comparatively higher in mucous cells as compared to other types of epithelial cells [21]. Alveolar type 1 (AT1) and alveolar type 2 (AT2) cells are susceptible to SARS-CoV-2 infection, with AT2 cells being the initial target [22]. Nonetheless, academic studies propose that a minor fraction of AT2 cells exhibit the expression of ACE2, while the expression of TMPRSS2 is minimal in the basal cells of the undifferentiated airway epithelium and more prominently expressed in the differentiated airway epithelium [23,24,25]. This proposed mechanism is accountable for the susceptibility of the lungs to COVID-19.

Dyspnea is the most common pulmonary symptom attributed to Long COVID. Dyspnea is defined as an unpleasant and unmanageable pattern of breathing related to an inability to ventilate well enough to provide the required amount of air for one normal breath. According to Hentsch L. et al., in their research published in 2021, three possible mechanisms were proposed for dyspnea induced by COVID-19 [26]. One of the mechanisms implies the interruption of afferent sensory signaling pathways by SARS-CoV-2, which may result in the inability of cortical structures responsible for processing the sensory aspects of dyspnea to receive afferent inputs from the brainstem. It is plausible that the virus may cause direct harm to the mechano- or irritant receptors located in the respiratory tract and/or chest wall, thereby impeding the transmission of afferent signals to the brainstem and higher brain structures. The second mechanism pertains to the possibility that SARS-CoV-2 may impede the capacity of cortical structures to identify or manage incoming sensory signals related to breathlessness originating from the brainstem. The occurrence of the conditions may arise either because of the direct impact of SARS-CoV-2 on nervous tissue or indirectly through the manifestation of inflammatory acute encephalopathy or cerebrovascular complications, including but not limited to ischemic or hemorrhagic stroke. The last mechanism implies that the cortical structures involved in the perception of dyspnea may exhibit a facilitative influence on the perception of breathlessness, like that of pain, which may be disrupted by SARS-CoV-2. The induction of dyspnea-related panic attacks through experimental inhalation of 35% CO_2_ has been demonstrated in patients with bilateral amygdala damage [27]. The self-reported levels of panic and fear in this group were found to be remarkably higher than those acknowledged by the comparison group with intact neurological function. The findings of this study may indicate that the activation of extra-limbic brain structures by CO_2_ occurs directly, and subsequently, these structures are controlled in a downward manner by the amygdala [27].

Several mechanisms of pulmonary damage in COVID-19 have been identified, with viral and immune-mediated pathways being implicated. Pulmonary fibrosis may arise because of chronic inflammation or as an idiopathic, age-related fibroproliferative process that is influenced by genetic factors [28]. Pulmonary fibrosis is a recognized consequence of acute respiratory distress syndrome (ARDS). Nevertheless, the clinical relevance of persistent radiological abnormalities after ARDS is limited and has decreased with the implementation of protective lung ventilation techniques [29]. Research has indicated that a significant proportion of individuals diagnosed with COVID-19, specifically 40%, are prone to developing ARDS. Furthermore, it has been observed that 20% of ARDS cases are classified as severe [30]. The manifestation of post-COVID-19 fibrosis will require further observation; however, initial examination of patients with COVID-19 upon hospital release indicates that over 33% of cured individuals exhibit fibrotic irregularities.

The defining characteristic of ARDS is the presence of diffuse alveolar damage (DAD). This is marked by an initial phase of acute inflammatory exudation, which is characterized by the presence of hyaline membranes. This is then followed by an organizing phase and a fibrotic phase [31]. Prior research has emphasized the significance of the duration of illness as a crucial factor in the development of pulmonary fibrosis after ARDS. The findings of this investigation indicate that a small proportion of patients (4%) who had a disease duration of less than one week, a notably larger proportion (24%) of patients with an illness lasting between 1 and 3 weeks, and a majority (61%) of patients who had a disease duration exceeding three weeks experienced the development of fibrosis [32]. The development and progression of pulmonary fibrosis may be instigated and facilitated by a cytokine storm resulting from an atypical immune response. The release of matrix metalloproteinases during the inflammatory phase of ARDS leads to epithelial and endothelial injury. The process of fibrosis involves the participation of vascular endothelial growth factor and cytokines, including interleukin (IL) 6 and tumor necrosis factor (TNF) a.

The etiology behind the differential outcomes of individuals who either recovered from an insult or developed progressive pulmonary fibrosis characterized by the accumulation of fibroblasts and myofibroblasts, along with excessive collagen deposition, remains unclear [33]. While COVID-19-induced ARDS appears to be the primary indicator of pulmonary fibrosis, various studies have indicated that it differs from classical ARDS in terms of its high and low elastance types.

The computed tomography (CT) results of numerous cases of COVID-19 do not indicate classical ARDS. In addition, the presence of abnormal coagulopathy is a notable pathological characteristic of this disorder. The mechanism underlying pulmonary fibrosis in COVID-19 differs from that observed in other fibrotic lung diseases, such as idiopathic pulmonary fibrosis (IPF). Notably, pathological observations suggest that the site of injury in COVID-19-induced pulmonary fibrosis is primarily the alveolar epithelial cells rather than the endothelial cells.

The act of coughing is a reflexive action that requires minimal conscious control. This reflex is initiated by the activation of peripheral sensory nerves that transmit signals to the vagus nerves. These nerves, in turn, provide sensory input to the brainstem at the solitary nucleus and the spinal trigeminal nucleus. The phenomenon of cough hypersensitivity has been established in the context of chronic cough, wherein the pathways responsible for coughing are believed to have undergone sensitization due to an increase in the magnitude of afferent signals transmitted to the brainstem. Coronaviruses, including SARS-CoV-2, gain access to host cells through specific receptors and proteases, namely ACE2, TMPRSS2, and furin [34]. SARS-CoV-2 may have the ability to directly interact with sensory neurons, as evidenced by the prevalence of sensory dysfunction, such as coughing, as well as olfactory and taste impairments, among individuals who have been infected with the virus [35]. The expression of ACE2 or TMPRSS2 in human airway vagal sensory neurons and their susceptibility to SARS-CoV-2 infection remain unknown. The bronchopulmonary vagal sensory neurons in mice were subjected to single-cell sequencing, which revealed the absence of murine ACE2 expression [36].

The potential involvement of supplementary viral entry factors in the interplay between SARS-CoV-2 and neurons cannot be disregarded. One such factor is neuropilin-1, which is present in vagal and other sensory neurons [37]. The study by D. H. Brann et al. conducted a sequencing analysis of human olfactory mucosal cells, revealing the absence of ACE2 and TMPRSS2 in olfactory epithelial neurons [38]. However, a significant expression of these genes was observed in support cells of the olfactory epithelium and stem cells [38]. The veracity of the results was validated through the cellular histological localization of ACE2 in the specialized neuroepithelium of supporting cells surrounding neuronal dendritic projections. It is noteworthy that the neuroepithelium comprises odor-sensing cilia [39]. Hence, it is plausible that the onset of anosmia resulting from SARS-CoV-2 infection could be attributed to the impact of the infected epithelium on neuronal function.

The ACE2 gene has been identified in a specific group of sensory neurons located in the thoracic ganglia of humans. These neurons are also known to provide innervation to the lungs. It is worth noting that a particular group of nociceptive neurons, which express calcitonin-related polypeptide alpha (CALCA) or purinergic receptor P2X 3 (P2RX3), have been found to exhibit expression [40]. These neuronal subtypes are like those found in the vagal sensory ganglia, which play an important role in triggering coughing. The similarity in developmental lineage and molecular phenotype between certain vagal sensory neurons, particularly those implicated in cough and dorsal root ganglion neurons, suggests a potential correlation between ACE2 expression in human vagal sensory neurons.

The etiology of persistent cough following SARS-CoV-2 infection is presently unknown, despite the possibility that the involvement of dorsal root ganglion neurons that contain nociceptors could account for the joint and chest pain, headache, and dyspnea symptoms experienced with Long COVID. The S1 spike protein of SARS-CoV-2 can cross the blood–brain barrier (BBB) in mice through absorptive transcytosis, indicating that a functional virus is not necessary for brain involvement [41]. Additional research is required to explore the potential direct interactions between the virus and the nervous system in the development of cough and other sensory symptoms in individuals with SARS-CoV-2 infection.

The exclusion of pathological or structural causes is crucial in the clinical management of chronic cough following COVID-19. This includes assessing fibrosis damage to lung parenchyma or damage to the airways resulting from SARS-CoV-2 infection or critical care management [42]. The presence of lung parenchymal alterations is a frequent observation on computed tomography (CT) scans in adult individuals affected by COVID-19. Additionally, a proportion of 10–20% of patients may experience the development of lung fibrotic changes [42]. The presence of lung fibrosis has been found to potentially heighten the sensitivity of the cough reflex in reaction to mechanical stimuli applied to the chest wall, as evidenced in individuals diagnosed with idiopathic pulmonary fibrosis [43].

The presence of a persistent cough in individuals experiencing post-COVID symptoms may be attributed to neuroinflammation, resulting in a state of heightened laryngeal and cough hypersensitivity. This phenomenon serves as the underlying cause of chronic refractory or unexplained cough [44,45]. Neuromodulators such as gabapentin and pregabalin have demonstrated efficacy in managing chronic refractory cough. The aforementioned strategy could be deemed as a viable option for addressing Long COVID, as these pharmaceutical agents may have utility in mitigating additional symptoms that coincide with coughing, such as discomfort, albeit with the possibility of exacerbating any cognitive impairment. The pulmonary manifestations of Long COVID are summarized in Figure 2.

### 3.2. Endocrinological Involvement in Long COVID

Endocrinologic sequelae of SARS-CoV-2 are caused by physiopathological changes specific to the virus, inflammatory and immunological lesions, iatrogenic complications, deficiency of vitamin D, and corticosteroid therapy. The ACE2 receptor is the key component through which SARS-CoV-2 obtains access to host cells. The spike glycoprotein of the virus is homotrimeric in nature and comprises S1 and S2 subunits. It extends outwards from the virus surface and plays a crucial role in binding to the ACE2 receptor [46,47]. Upon binding to ACE2, the S1 subunit is dissociated with the ACE2 receptor in a process that requires the presence of TMPRSS2 [48]. Multiple endocrine tissues express abundant ACE2 and TMPRSS2, including the hypothalamus, pituitary, thyroid, adrenal, gonads, and pancreatic islets, which underlines the necessity to investigate further endocrine post-COVID manifestations related to these structures [49].

#### 3.2.1. Pituitary Gland

Several case reports have indicated that there may be a high risk of pituitary apoplexy among patients with pituitary neoplasms and COVID-19 infection. While some of these studies included additional risk factors for apoplexy, such as pregnancy, and the majority involved patients with pre-existing pituitary macroadenomas, others involved patients with microadenomas, which are typically associated with apoplexy less frequently [50,51,52].

#### 3.2.2. Thyroid Gland

Thyroid tissue demonstrates a significant level of ACE2 messenger (m) RNA expression, and regardless of gender, it expresses a significant level of TMPRSS2 [49,53]. Moreover, the thyroid’s location near the upper airway offers the virus an entry point to access the thyroid gland [54].

Reduced levels of thyroid-stimulating hormone (TSH) are accompanied by an increase in viral load in COVID-19 patients. IL-6 and TNF-a, which are pro-inflammatory cytokines, and increased ACE2 expression in the vascular endothelium are the underlying mechanisms of post-COVID thyroid disturbances. COVID-19 thyroid involvement can manifest as sick euthyroid syndrome, subacute thyroiditis (SAT), Graves’ disease, postpartum thyroiditis, Hashimoto’s disease, and silent thyroiditis [54,55,56]. SARS-CoV-2 may catalyze SAT. In a study conducted by Fatourechi V. et al., among 38 patients who developed SAT following COVID, the onset of the complication was reported days, weeks, or even months after the initial viral infection, exhibiting symptoms identical to typical cases of SAT, with most experiencing mild forms of COVID-19. Four cases of hypothyroidism were reported post-SAT, and there were no reported deaths [57].

#### 3.2.3. Pancreas

The pancreas can be affected by COVID-19 through different mechanisms, including direct viral injury, systemic inflammation, lipotoxicity, and drug-induced pancreatitis [58]. Some cases of necrotizing pancreatitis and acute pancreatitis are considered long-term COVID complications [59]. Many patients experience impaired glucose tolerance and insulin resistance, which may lead to type 2 diabetes (T2DM). Long-term COVID symptoms include various endocrine and metabolic conditions, such as newly diagnosed diabetes, and severe metabolic complications like diabetic ketoacidosis [60]. The relationship between COVID-19 and diabetes appears to be bidirectional, with pre-existing diabetes increasing the risk of severe COVID-19 and COVID-19 infection potentially causing new-onset diabetes during the Long COVID course [61]. SARS-CoV-2’s diabetogenic effects are significant, as shown by findings of hyperglycemia, severe insulin resistance, increased ketosis, and hyperosmolar hyperglycemic states in older individuals. These effects go beyond the usual stress response associated with serious illnesses [60]. The possible mechanism by which SARS-CoV-2 triggers T2DM involves morphological changes in pancreatic islet remodeling, including the activation of the renin–angiotensin–aldosterone system (RAAS), islet redox, inflammation, amyloid formation, fibrosis, and beta cell dysfunction or loss, along with capillary rarefaction. The virus directly affects beta cells, leading to their malfunction, failure, and apoptosis, while the cytokine storm indirectly accelerate islet remodeling in the progression of T2DM [62].

Specifically, after the activation of the systemic and islet RAAS, various events occur before angiotensin II formation [63], a potent vasoconstrictor that, when present for a long time or in large quantities, could contribute to the development of T2DM. Similarly, SARS-CoV-2 activates this system, increasing the potential for Long COVID T2DM. Additionally, the binding of SARS-CoV-2 to the ACE2 receptor may impact the pancreatic islet-exocrine interface through the involvement of exocrine pancreatic ductal cells [64]. Wang F. et al. assert in a study conducted on 52 COVID-19 patients with pneumonia that pancreatic injuries are present in approximately 17% of them [65]. Consequently, SARS-CoV-2 may affect both the endocrine and exocrine tissues of the pancreas during COVID-19 infections [65].

#### 3.2.4. Adrenal Glands

Long COVID symptoms were reported to be like low cortisol symptoms [66,67]. Recent research indicates that asthenia, myalgia, and arthralgia influence 65.5%, 50.6%, and 54.7% of Long COVID patients, respectively. This could result from the suppression of the hypothalamic–pituitary–adrenal axis caused by high doses of dexamethasone and direct free radical stress [68]. Furthermore, recent studies have shown a similarity in specific amino acid sequences between adrenocorticotropic hormone (ACTH) and SARS-CoV-2. This similarity has led researchers to theorize that COVID-19 infection might stimulate the creation of antibodies that react with both, potentially deactivating natural ACTH and eliminating cells that produce ACTH [69].

Hypocortisolism following the COVID-19 infection was also associated with adrenal cortex impairment. ACE2 expression was detected in the zona fasciculata and zona reticularis of the adrenal cortex, suggesting that a COVID-19-induced adrenal tissue injury might affect the production of glucocorticoids [62]. Autopsies conducted on COVID-19 patients revealed necrosis of the adrenal cortical cells and the presence of the virus at this level [70].

#### 3.2.5. Gonads

It is believed that SARS-CoV-2 affects both testis cells and sperm due to increased expression of ACE2 receptors on both Sertoli and Leydig cells and in sperm, leading to hypogonadism [68]. Postmortem histopathological examination of the testes of COVID-19 patients revealed Sertoli cell enlargement, vacuolization, and cytoplasmic thinning; tubule detachment from basement membranes; and a decreased number of Leydig cells [69]. In addition, SARS-CoV-2 infection associated with hypogonadism and impaired endothelial function contributes to erectile dysfunction. The prevalence of erectile dysfunction was found to be higher in individuals who had previously had COVID-19 [70].

Ovarian tissues, as well as the uterus, placenta, vagina, and breasts, express ACE2 receptors. However, SARS-CoV-2 does not appear to infect oocytes [71]. Infection with SARS-CoV-2 was associated with a change in menstrual volume and the duration of the menstrual cycle, regardless of the severity of the infection, according to the findings of a retrospective study [71]. In an international cohort study of women with Long COVID syndrome, menstrual disturbances, particularly irregular menstruation and unusually heavy periods/clots, were also observed [72]. The long-term effects of SARS-CoV-2 infection on the human endocrine system are summarized in Table 1.

### 3.3. Hematological and Vascular Mechanism of Long COVID

Long after the original infection, many hematological and vascular manifestations continue to persist [73]. Relapsing-remitting symptoms may be seen in long-term COVID patients [74]. Some of the most studied hematological and vascular SARS-CoV-2 mechanisms that can cause recurrent multi-organ damage are cell count alterations, viral persistence in the extracellular vesicles (EVs), the hypercoagulable state, the active vascular inflammatory state, endothelial damage, and microangiopathy [75].

#### 3.3.1. Cell Count Alterations

The hematological consequences of COVID-19 are of serious concern, as they present significant lymphopenia and cluster of differentiation (CD) 8+ T-cell dysfunction, which eventually lead to a cytokine storm and a compromised immune system [76]. There is a considerable increase in mortality risk associated with COVID-induced neutrophilia, which is most likely in the context of bacterial superinfection with opportunistic pathogens, all of which are particularly concerning [77]. Up to 75% of severely sick COVID-19 patients in intensive care units (ICUs) have been found to have neutropenia and lymphopenia with the persistence of these blood counts for more than 6 weeks after discharge, and those who passed away had lymphocyte counts that were continuously declining by the time of death [77,78].

Low levels of thrombocytes are correlated with the intensity of the disease, and several analyses reveal that a low level of platelets is linked to a significantly greater chance of developing severe COVID-19, with an almost five-fold higher likelihood [76,78]. The primary mechanisms believed to contribute to the development of thrombocytopenia in COVID-19 include direct bone marrow toxicity resulting from the viral infection, endothelial damage induced by assisted ventilation, and aberrant platelet activation [76,78].

#### 3.3.2. SARS-CoV-2 Persistence in Extracellular Vesicles

Various cell types release extracellular vesicles containing mRNA, microRNAs, deoxyribonucleic acid (DNA), lipids, and proteins. These EVs serve the purpose of facilitating the maintenance of the physiological state of neighboring or distant cells. Recent investigations have suggested that SARS-CoV-2 has the potential to be transmitted to distant tissues and organs via EVs [73,79,80,81,82,83,84,85,86,87,88,89,90].

Additionally, numerous studies have demonstrated that EVs are crucial for the activation of coagulation. Chronic hypoxia brought on by Long COVID is frequently accompanied by pulmonary vascular abnormalities and impaired lung function. Additionally, hypoxia creates conditions for immune cells to release more pro-inflammatory cytokines [90,91]. Exosomes, microparticles (MPs), and apoptotic bodies are the three forms of EVs. Because of their small size, biogenesis technique, and cell entry mechanism, EVs and viruses share many structural similarities. Most enveloped RNA viruses break out of their host cells or from the plasma membrane by budding. The biosynthetic secretory pathway facilitates the release of identical SARS-CoV-2 particles from the endoplasmic reticulum (ER)-Golgi intermediate compartment (ERGIC) or Golgi apparatus into the external environment of the cells. According to some research, SARS-CoV-2 can exit cells as tiny secretory vesicles that later release the virus [92].

Most enveloped RNA viruses break out of their hosts’ cells or from the plasma membrane by budding. SARS-CoV-2 RNA was discovered in exosomal cargo in other experiments, raising the possibility that the virus might spread infection via the exocytic pathway [92]. This shows that one potential mechanism for the recurrence of the COVID-19 infection may be the cellular transport pathway linked to the release of EVs containing SARS-CoV-2. EVs might act as a “trojan horse” for the viral RNA to resurface in COVID-19 patients who have recovered. SARS-CoV-2 has the potential to reside within particles outside the cells during cases of long-term COVID-19, subsequently re-engaging with various tissues and organs through the bloodstream [83,93,94]. In the end, persistent endothelial damage, widespread vascular endotheliitis, and thrombosis are caused by persistent viral presence, hypoxia, and inflammatory reactions [92].

#### 3.3.3. Persistent Vascular Inflammatory State

The interaction between SARS-CoV-2 and the humoral innate immune system, more specifically the interaction between the coagulation, fibrinolytic, and complement systems, is what primarily causes thrombo-inflammation. Endothelial cell (EC) activation and/or injury, leukocytes, and platelets all lead to thrombotic and inflammatory reactions [73]. Thromboembolic complications affecting both the venous and arterial circulation could result from the ongoing endothelial dysfunction that occurs during Long COVID in combination with an impaired innate immune response. In addition to causing excessive thrombin production, fibrinolysis inhibition, and ongoing activation of the complement pathways, SARS-CoV-2-mediated endothelial dysfunction can further extend the microvascular system’s dysfunction and cause microthrombosis [89]. Acute COVID-19 can become more severe due to a cytokine storm. It was shown that those who went on to acquire Long COVID typically had higher levels of cytokine biomarkers during the early stages of recovery, including TNF, interferon (IFN)-induced protein 10, and IL-6, associated with enhanced immune activation. According to some theories, chronic immune activation is caused by persistent viral RNA discharge. Elevated levels of IFN gamma and IL-2, pathological irregularities in CD4+ and CD8+ lymphocyte groups, as well as in monocyte CD14+ and CD16+ groups, along with deficiencies in B lymphocytes and monocytes, have been identified as indicative of immune system dysregulation in chronic COVID [94].

The advancement of COVID-19 is driven by heightened oxidative phosphorylation, inflammatory responses associated with reactive oxygen species, and the displacement of inflammatory responses mediated by TNF and IL-6. Therefore, long-term, continuous chronic inflammation in COVID may activate endothelial cells (ECs), platelets, and other inflammatory cells, encourage the overexpression of procoagulant factors, and undermine the protective role of the vascular endothelium, leading to aberrant coagulation. Due to these consequences, there is a feedback loop whereby thrombosis leads to inflammation, and the ensuing blood clots may themselves cause further inflammation. An immediate connection between coagulation and inflammation is made possible by thrombin, which breaks down fibrinogen and stimulates IL-1 [94].

#### 3.3.4. Endothelial Damage and Dysfunction

Impaired endothelial function can serve as an independent risk factor for the development of Long COVID syndrome. In lengthy COVID, vascular endothelial damage is also typical. In recovering COVID-19 patients, EC biomarkers such as von Willebrand factor (vWF) antigen, vWF pro peptide (vWFpp), and factor VIII (FVIII: C) are markedly elevated [83]. The primary connection is between the mechanisms that encourage thrombosis and vascular endothelial damage. The entire vascular system is covered by ECs, which also regulate the circulation of blood and coagulation, generate and amplify inflammatory responses, and keep vascular tension, shape, and homeostasis in balance [93]. Via low levels of antithrombin III, tissue factor pathway inhibitor, and protein C, vascular endothelial damage weakens the cell’s anticoagulant capabilities and, at the same time, increases permeability and leukocyte adhesion. By increasing the expression of tissue factor (TF), exposing phosphatidylserine (PS), and releasing vWF and factor VIII, injured ECs become procoagulant. Additionally, ECs can enhance the recruitment of neutrophils, up-regulate chemokine production on their surface, and take part in thrombosis [94]. ECs complications brought on by inflammation may result in a sharp rise in plasminogen activator, correlated with the elevated D-dimer levels in individuals with severe COVID-19. Additionally, capillary leakage can be accelerated, and the extracellular matrix can be modified because of plasmin’s effects on metalloproteinases. As a result, organ malfunction and post-acute symptoms may also be caused by endothelial injury and chronic dysfunction [94].

In COVID-19, endothelium injury brought on by the virus, inflammation, and hypoxia may result in decreased flow rate and wall shear stress, which can lead to platelet aggregation and thrombosis. In addition, one of the signs of endothelial dysfunction seen in many organs of deceased COVID-19 patients is intussusception angiogenesis (IA). This is a quick angiogenesis procedure that uses circulating angiogenic cells to divide the blood artery into two lumens. Important causes of IA include hypoxia, traditional angiogenic molecular factors, severe inflammation and cytokine storms, thrombosis, associated hemodynamic abnormalities, and dysfunction of RAAS products. Physiological laminar flow may also be hindered by the vascular control problems seen in focal vasoconstriction and progressively dilated artery segments [94].

#### 3.3.5. Long-COVID-Related Hypercoagulability

One of the mechanisms of the hypercoagulation state is the fact that endothelial damage is prone to evolving into endotheliitis, which increases the risk of hypercoagulability. The Long COVID patient is more likely to experience acute thrombotic events, which in turn trigger acute stroke, venous thromboembolism, disseminated intravascular coagulation (DIC), and acute ischemic events. Numerous clinical research studies and trials have demonstrated the necessity of routine blood counts and inflammatory marker tests for Long COVID patients, namely for those who are older and weaker [75,95,96,97].

In addition to eliciting inflammatory immunological responses, SARS-CoV-2 has the capability to cause direct harm to ECs through its attachment to ACE2 receptors during the acute phase of COVID-19. The consequences of endothelial cell damage include endothelial activation, impairment of barrier function, heightened permeability, and leakage in the microvasculature [79]. EC’s damage can lead to systemic and widespread endothelial dysfunction, as well as the activation of numerous immune-mediated inflammatory pathways and thrombus formation, all of which can cause serious multi-organ involvement and subsequently higher mortality [80]. SARS-CoV-2 has also been discovered in ECs, and patients hospitalized for COVID-19 have higher than average levels of circulating ECs, indicative of vascular damage. However, even after SARS-CoV-2 is not detected anymore, some pathogenic processes continue [81].

A large percentage of recovering patients had elevated markers of endothelial lesions and coagulation, which suggests infection may cause persistent coagulopathy, endotheliitis, and microvascular lesions, along with the creation of microthrombi [82]. Persistent endothelium lesions are frequent in the COVID-19 convalescent period, according to study findings overall. By producing antiplatelet and anticoagulant chemicals to prevent the development of fibrin clots or platelet aggregation and inhibiting the binding of clotting proteins, the normal endothelium plays a critical role in avoiding thrombosis. It has been established in the past that endothelial dysfunction raises the risk of thrombosis. Therefore, endothelial dysfunction because of endothelial damage may be a secondary cause of Long COVID thrombosis [83]. Anticoagulation has taken a prominent role in the complete care of COVID-19 infection, as hypercoagulability is becoming a more widely recognized consequence of COVID-19 infection [94,98].

The pathophysiology of both COVID-19 infection and its persistent phenotype, known as Long COVID, is believed to be influenced mostly by the presence of abnormal microclot formation, specifically amyloid microclots, which are resistant to fibrinolysis. Additionally, there is an increase in 2-antiplasmin (2AP) levels and a surge of acute-phase inflammatory molecules. In a 2021 study conducted by Pretorius et al., an analysis of samples from COVID patients and those with Post-Acute Sequelae of COVID-19 (PASC), which is associated with persistent clotting issues in Long COVID, revealed notable increases in dysregulated molecules between acute COVID-19 cases and long-term COVID [87]. The prolonged inflammatory status in Long COVID/PASC and persistent viral infection were considered possible factors.

The study emphasizes the need for larger sample sets to validate these proteomics findings due to the limited statistical power of the current collection. Additionally, platelet aggregation assays, prothrombin time, and partial thromboplastin time analysis should be considered. The study concludes that circulating microclots and hyperactivated platelets, hypercoagulability driven by significant increases in inflammatory markers, and an aberrant fibrinolytic system result from dysregulated clotting proteins and lytic enzymes, with a substantial rise in 2AP playing a crucial role. Consequently, ongoing anticoagulant therapy could potentially benefit both acute COVID-19 infections and long-term COVID/PASC patients with clotting disorders.

#### 3.3.6. Microangiopathy

Both initial COVID and Long COVID are characterized by coagulopathies and the development of microclots in vivo [84]. Clinical research and studies have shown that these microclots are also amyloid in nature. The production of abnormal clots that take amyloid states and do not undergo fibrinolysis can be induced by adding pure, recombinant SARS-CoV-2 S1 spike protein to normal plasma [85]. It should be noted that exogenous thrombin was not used in these cases. It has been demonstrated that these clots are identifiable within the bloodstream of individuals experiencing both acute and chronic COVID-19 [86,87]. These Long-COVID-associated microclots are resistant to fibrinolysis; they contain antitrypsin, making them resistant to lysis processes; they contain inflammatory molecules, which may cause inflammation upon lysis; and they are hypothesized to obstruct the microcirculation in different capillary beds, causing tissue dysfunction and eventually leading to organ dysregulation and dysfunction, which results in clinical presentations relating to an organ [88].

In a recent study provided by Pretorius E. et al., it was demonstrated that plasma from Long COVID patients had microclots that were resistant to fibrinolysis [87]. Platelets have been reported to contain SARS-CoV-2 RNA, and severe acute infection is associated with platelet activation and degranulation. It is still unclear exactly how platelet activation caused by SARS-CoV-2 viral contacts works. However, several mechanisms might be able to account for this phenomenon. First, by directly interacting with the host platelet receptor, the viral spike protein of SARS-CoV-2 can connect with platelets and perhaps activate them. Indeed, the SARS-CoV-2 spike protein binds to the highly abundant ACE2 receptors on platelets with strong affinity. When coated with antiviral antibodies capable of attaching to the human immunoglobulin receptor IIa (FcRIIa) CD 32a on platelets, SARS-CoV-2 may also bind to them [99].

The arginine–glycine–aspartic amino acid sequence found in the SARS-CoV-2 spike protein can interact with integrins and bind to CD 209 on platelets. The main receptor on the surface of platelets is glycoprotein IIb/IIIa, which can quickly bind a variety of ligands that contain the arginine–glycine–aspartic acid sequence. It has also been demonstrated that the SARS-CoV-2 envelope protein interacts with the Toll-like receptor (TLR) 2 present on platelets. Platelet aggregation and the creation of three-dimensional clots are caused by the release of thrombin (IIa) by activated platelets. This process is triggered by platelet adhesion receptor glycoprotein (GP) Ib-IX-V, which is found on the surface of platelets, binding to collagen and vWF that are exposed to injured artery walls [99].

### 3.4. Gastroenterological Involvement in Long COVID

Gastrointestinal (GI) manifestations are one of the main organ-specific complications of acute and Long COVID, and this is believed to be due to the ACE2-expressing cells on the brush border of the small intestinal mucosa [100,101,102,103,104,105,106,107,108,109]. The GI symptoms displayed during acute COVID-19 differ from those displayed in post-COVID-19 syndromes. The acute symptoms are predominantly abdominal pain, nausea, vomiting, diarrhea, and abnormal liver function tests [105,108,110,111]. The post-COVID-19 syndrome manifestations present as loss of appetite, irritable bowel syndrome, weight loss, altered bowel motility, and dysphagia [101,108]. The presence of GI symptoms during the acute stage of the infection is linked to a higher likelihood of long-term GI post-COVID-19 problems [107]. As surmised in the systematic review by Choudhury et al., GI symptoms were reported in 14 out of the 50 studies conducted [103]. The 14 studies involved 296,487 patients, and the frequency with which patients with COVID-19/Long COVID experienced GI manifestation was 0.12/0.22 [103]. The relationship between Long COVID and inflammation has been evaluated in recent studies. Disruption of the gut-lung axis, an established indicator of illness severity in other respiratory conditions, may play a role [112]. Direct or indirect effects of SARS-CoV-2 on the GI tract can cause breakdowns in the integrity of the gut barrier, allowing gut bacteria and their products to translocate across the gut epithelium and escalating early systemic inflammation [112,113,114,115,116,117,118,119,120,121].

Acute COVID-19 has also been linked to elevated plasma levels of zonulin, a measure of tight junction permeability that promotes microbial translocation and inflammation [112]. There have also been reports of persistent shedding of SARS-CoV-2 in the feces and the existence of the virions in intestinal cells (enterocytes) [102,114,115]. This supports the hypothesis that the virus infects the GI tract through the fecal–oral route [115,116,117,118,119,120,121]. Gaebler C. et al. demonstrated that the antigen persisted in the intestinal tract even after it had been cleared from the nasopharynx and the clinical illness resolved [102]. Of the biopsies obtained from the GI tracts of 14 former COVID-19-positive patients, 5 were found to have the SARS-CoV-2 protein in the enterocytes.

A study conducted in Texas aimed to evaluate the long-term sequelae of COVID-19 in patients with no associated pathology [104]. Of the 101 patients present at the COVID-19 recovery clinic, 25 patients had no comorbidities. As to be expected, almost all the patients reported respiratory symptoms (dyspnea and cough) as well as persistent GI manifestations, e.g., abdominal pain, nausea, diarrhea, and vomiting in 84% of the patients. The study concluded that although comorbidities are a risk factor for severe COVID-19 symptoms, having no comorbidities does not protect against prolonged recovery [101].

A study conducted by Suárez-Fariñas M. et al. in 2021 set out to determine if COVID-19 influences inflammatory bowel disease (IBD) and IBD medication and if there is a shared pathological pathway between them [100]. The study found that genes associated with the inflammatory response in IBD overlap with genes that respond to COVID-19 infection. Through examining the blood of patients infected with COVID-19, it was observed that there was a higher expression of upregulated genes in the patients with IBD than there was in healthy control blood. The studies concluded that there is potential for local, GI-associated replication of the SARS-CoV-2 because of the high expression of ACE2 and TMPRSS2.

### 3.5. Cardiovascular Involvement in Long COVID

The evaluation of prolonged cardiovascular issues is ongoing, and the understanding of the actual impact of post-COVID-19 cardiovascular complications remains uncertain. COVID-19 appears to worsen pre-existing cardiac pathology but also causes new-onset cardiac diseases such as arrhythmias or hypertension (Figure 3).

While the causes behind enduring cardiac injury post-recovery are not yet fully grasped, a potential scenario points to a persistent inflammatory reaction reflected by the presence of circulating biomarkers for 3 to 8 months after acute infection: C-reactive protein, IL 6 and 8, ferritin, IFN beta and gamma1, procalcitonin (correlated with microvessel disease), chemokine ligand (CXCL) 9 (small cytokines that induce chemotaxis, promote differentiation and multiplication of leukocytes, and cause tissue extravasation), CXCL10 (an ‘inflammatory’ chemokine), T-cell immunoglobulin mucin-3 (TIM-3) (involved in immune response), plasma ACE2 activity, and pentraxin 3 (PTX3) (activates complement and facilitates pathogen recognition by macrophages). Some of these biomarkers were also found elevated in asymptomatic post-COVID patients during the period lasting up to 8 months [122]. This chronic inflammatory status may be caused by persistent viral reservoirs in the intestines or other immune-privileged sites following the acute infection [102].

Immune exhaustion after prolonged antigen stimulation or immunosuppressive treatment may also facilitate the development of cardiac injury. Within the cardiac structure, SARS-CoV-2 primarily resides within interstitial cells and infiltrating macrophages within the myocardium. Viral entry depends upon the interaction between viral spike glycoprotein, the ACE2 receptor, a transmembrane aminopeptidase in the host cell membrane, and the host cell protease system, such as TMPRSS2. ACE2 plays a crucial role in the neurohumoral regulation of the cardiovascular system and is primarily present in the vascular endothelium, cardiomyocytes, pericytes, cardiac fibroblasts, and epicardial adipose tissue. ACE2 naturally transforms angiotensin 1 and 2 into beneficial peptides like angiotensin 1–7 and angiotensin 1–9, which hold cardioprotective properties. SARS-CoV-2 causes downregulation of ACE2, thus preventing the synthesis of cardioprotective peptides and predisposing to cardiovascular damage.

Another proposed mechanism of myocardial damage is mediated by reactive oxygen species, which can induce the release of internal histones, damage-associated molecular patterns (DAMPs), and oxidized lipid–protein complexes. During the post-acute phase, these compounds trigger an inflammatory response that results in substantial myocardial tissue injury, leading to chronic myocardial scarring. This scarring may cause issues like reduced ventricular compliance, impaired blood flow within the heart muscle, decreased cardiac muscle contraction, and potential irregular heartbeats. Cytokine-induced damage can also manifest as blood clot formation, reduced oxygen delivery, destabilization of coronary plaques, the transition of chronic heart conditions into unstable states, increased metabolic requirements, diminished cardiac capacity, and inflammation in the heart valves [122].

A growing number of reports on cardiovascular autonomic dysfunction (CVAD) in PASC patients, namely patients who experience symptoms suggestive of postural orthostatic tachycardia syndrome (POTS), with an exaggerated rise in heart rate and intolerance to standing, commonly affecting females and individuals aged between 15 and 45 years old.

Infections commonly trigger dysautonomia, and several potential ways in which SARS-CoV-2 might contribute to this condition have been suggested based on initial evidence: hypovolemia, brainstem involvement, and autoimmunity [102,123]. Hypovolemia is a recognized characteristic in patients with PASC, leading to a hyperadrenergic response. This reaction subsequently leads to cerebral hypoperfusion and disruption of central autonomic networks. Brainstem dysfunction encompasses various mechanisms, including direct invasion by the virus, neuroinflammation, brainstem compression, and vascular activation. Autoimmunity has an important role in the pathophysiology of post-viral POTS, as supported by recent evidence [124]. A high prevalence of specific autoantibodies has been found in the sera of PASC patients presenting dysautonomia. They include G-protein coupled receptor (GPCR) antibodies, which can activate adrenergic receptors and elicit a negative allosteric effect on muscarinic GPCRs. Autoantibodies can also activate cholinergic receptors and cause peripheral vasodilation. Patients experiencing POTS linked to PASC have exhibited the presence of serum anti-nuclear, anti-thyroid, anti-cardiac protein, anti-phospholipid, and Sjogren’s antibodies [125]. Alarmingly, a possible shift in the age distribution of Long COVID can be expected due to an increased number of young, infected, unvaccinated individuals, whose morbidity and mortality are difficult to predict, according to the European Society of Cardiology [126].

### 3.6. Renal Involvement in Long COVID

Impairment of the kidneys in individuals who contracted COVID-19 has been widespread worldwide and manifests as hematuria, proteinuria, acute kidney injury (AKI), end-stage kidney disease (ESKD), and major adverse kidney events (MAKE) [127,128,129,130,131]. Even in the absence of a coexisting elevation in serum creatinine concentration or a decline in estimated glomerular filtration rate (eGFR), the presence of low molecular weight proteinuria or hematuria indicates a subclinical AKI. The mechanisms behind these injuries are multiple and include direct injury to the renal cells, indirect injury through other organ destruction, etc.

The ACE2 receptor is expressed in the epithelium of the kidneys and plays a major role in the entrance of SARS-CoV-2. Previous studies have shown that ACE2 is the functional receptor for SARS-CoV in vivo and in vitro. Recent cryo-electron microscopy has shown that the spike protein of SARS-CoV-2 binds directly to ACE2 with an even greater affinity than SARS-CoV [132].

Several studies have revealed that SARS-CoV-2 relies on the widely expressed transmembrane glycoprotein CD147 as a crucial element in infiltrating target cells [131,132]. CD147 is expressed in high amounts on the cell surface of the proximal tubular epithelial cells and is believed to have contributed to the pathogenesis of several renal diseases through its involvement in the immune-inflammatory response and dysregulated cell cycle [131]. It is established that the CD147 partners, cyclophilins, have a crucial role in the replication process of SARS-CoV-2, and inhibiting these by cyclosporins has demonstrated effective suppression of intracellular viral propagation [130]. Su H. et al., in their autopsy results of patients with COVID-19, demonstrated by electron microscopic examination agglomerates of coronavirus-like particles having distinctive spikes within the tubular epithelium and podocytes [130]. They further described notable tubular impairment, especially within the early segment of the proximal tubule, marked by an absence of microvilli, acute tubular necrosis, a notable decrease in megalin expression within the microvilli, and the existence of intraluminal debris. All denoting the presence of AKI [130]. Further findings of the virus in urine only strengthen this discovery [133]. Diao B. et al. further cemented this discovery when they confirmed the findings of SARS-CoV-2 in renal cells, further demonstrating that SARS-CoV-2 can replicate in vivo [134].

A study conducted by Chand S. et al. on 300 survivors of critical COVID-19, conducted in 2022, found that out of the patients who survived 120 days after COVID resolution, 74.4% had AKI resolution, while 25.6% did not have a resolution. Out of the 25.6% with AKI resolution, 60% developed chronic kidney disease (CKD), and 40% required renal replacement therapy (RRT) [129].

According to Bowe B. et al., in their cohort study conducted for 1 year on 1,726,683 US Veterans and 89,216 patients who were 1-month survivors of SARS-CoV-2 infection, patients who successfully recovered from COVID-19 displayed kidney-related symptoms during the post-acute stage, such as eGFR decline, AKI, ESKD, or MAKE. These symptoms increased in the post-acute period with the severity of the acute infection, and the risk for renal problems was higher in hospitalized patients who developed AKI [128].

### 3.7. Dermatological Involvement in Long COVID

Dermatological manifestations and complications of COVID-19 are becoming increasingly acknowledged in the literature. There are many types of skin manifestations described in conjunction with COVID-19, such as maculopapular rash, urticaria, petechiae, purpura, vesicles, chilblains, livedo racemose, and ischemia of the distal segments of the limbs [135,136,137]. These dermatological findings are significant as they can help the clinician make a COVID-19 diagnosis. There are various hypothesized causes of rash in COVID-19 patients. Diffuse microvascular vasculitis, brought on by complement system activation, is the first. In one study carried out by Magro C. et al. in 2020, interstitial and perivascular neutrophilia with pronounced leukocytoclasia, considerable complement protein deposition in the dermal capillaries, and other findings point to vasculitis phenomena [135]. Other studies performed by Sanchez A. et al. in 2020 suggest that the rash occurs as a direct effect of the virus. The justification is the presence of lymphocytosis (without eosinophils), papillary dermal oedema, epidermal spongiosis, and lymph histiocytic infiltrates [136]. Five clinical patterns were identified in a recent study conducted in Spain that included 375 cases: maculopapular eruptions (47%), urticarial lesions (19%), various vesicular eruptions (9%), and livedo or necrosis (6%) [138]. Although the face is often spared, the lesions are mostly restricted to the trunk and extremities (hands and feet).

The average lesions last a few days, although others have been reported to persist for as short as 20 min or as long as four weeks. Nevertheless, in some individuals, lesions emerged 2 to 5 days before the beginning of COVID-19 symptomatology. In most cases, the mean latency, that is, the amount of time it took for skin lesions to arise following the onset of the first typical COVID-19 symptoms, was between 1 and 14 days [139]. There is still a lack of knowledge on the pathophysiological mechanisms of skin lesions in COVID-19 patients. The skin manifestations presented in COVID-19 may be divided into two categories regarding their pathological mechanisms [140]. The first is an immune response to the viral nucleotides of COVID-19, which clinically manifests in a similar way to viral exanthems. The second group includes skin eruptions that result from the systemic effects of COVID-19; this is especially the case in vasculitis and thrombotic vasculopathy.

The possible actions of SARS-CoV-2 on the skin may be mediated by non-structural viral proteins (NSPs) that block the innate immune system (NSP3, NSP16), inhibit the effect of IFN (NSP5), or synthesize cytokines (NP3). The cytokine storm can stimulate dermal cells, leading to the appearance of various maculopapular or vesicular rashes. Secondary activation of complement by the surface viral antigen and microangiopathy may lead to the appearance of purpuric manifestations [139,141]. TMPRSS2 activation is crucial for the virus to attach to ACE2 through its spike protein. Androgen receptor elements near the TMPRSS2 gene on chromosome 21 suggest COVID-19 could affect men more due to androgen levels. The higher COVID-19 rates in Spanish men, possibly linked to androgenetic alopecia prevalence, hint at androgens playing a role in worsening the severity of the disease [142]. Another possible effect of SARS-CoV-2 is a direct impact through ACE2 in the epidermis, causing certain skin conditions (acantholysis and dyskeratosis) [143]. ACE2 appears in the skin’s basal epidermal layer, dermal blood vessel ECs, and eccrine tissue [139,144]. The reduced microcirculatory function across arterial beds in COVID-19 patients and its effects may be linked to COVID-19-related endotheliitis involving ACE2 [139,140].

### 3.8. Neuropsychiatric Mechanism of Long COVID

Up to one-third of individuals with Long COVID, also known as PASC, may have ongoing neurological problems that manifest as anosmia, hypogeusia, lethargy, headaches, “brain fog”, dysautonomia, cognitive impairment, and peripheral neuropathy [145].

The neurological manifestations observed in cases of acute COVID-19 have been associated with various interconnected pathogenetic mechanisms. These mechanisms include viral invasion of the nervous system accompanied by abnormal immune responses, dysfunction of the blood–brain barrier due to epitheliopathy, coagulation disorders leading to neuronal injury caused by reduced oxygen supply, imbalances in metabolic processes, cascades of oxidative stress, and cellular apoptosis [146]. These factors are posited as potential causes for the enduring symptoms observed in individuals affected by COVID-19. SARS-CoV-2 can invade the stem and support cells located in the olfactory epithelium, which is situated outside the central nervous system. This invasion can lead to persistent alterations in the sense of smell. Anomalies in innate and adaptive immunity that may arise from SARS-CoV-2 infection include monocyte enlargement, T-cell exhaustion, and prolonged cytokine release [145]. These abnormalities may lead to neuroinflammatory reactions and microglia activation, abnormalities in the white matter, and microvascular alterations. Additionally, viral protease activity and complement activation can cause endotheliopathy, which can lead to hypoxic neuronal damage and BBB failure, and microvascular clot formation, which can obstruct capillaries [145].

#### 3.8.1. SARS-CoV-2 Neurotropism

It is now believed that SARS-CoV-2’s direct neurotropic actions have a limited impact on the etiology of Long COVID, like the part that invasion of the nervous system played in acute COVID-19 [147]. A detailed study by Meinhardt J. et al. proved the presence of SARS-CoV-2 in the olfactory cells [147]. The serine protease TMPRSS2 is known to act on the cellular receptor ACE2 and to bind to the SARS-CoV, especially SARS-CoV-2, allowing the virus to enter human host cells. It has been demonstrated that non-neuronal cells in the human olfactory mucosa physiologically express ACE2 [147]. It was determined that the olfactory mucosa had the highest concentration of S protein [147]. Using immunohistochemistry, unique immunoreactivity for the S protein was discovered in cell types that differ morphologically and are suggestive of neuronal/neural origin [147]. This immunoreactivity included a distinctive granular, partially perinuclear pattern. The RNA of SARS-CoV-2 was identified in cells of the olfactory epithelium and nasal mucus [147].

Coronaviruses and other neurotropic viruses employ sensory and motor neural pathways to penetrate the central nervous system (CNS). The olfactory nerve is a type of neural route. The way olfactory nerves are structured differently in the nasal cavity and forebrain, as well as the olfactory bulb, mediates this. Inflammation and a demyelinating reaction may result from the virus getting into the brain and cerebrospinal fluid (CSF) in this way. In less than 7 days, once the infection is established, the viruses can infect the entire brain and CSF [133,148]. Hematogenous spread from severely infected airways and lungs is one way that SARS-CoV-2 may enter the CNS. This sort of spread would be made easier by systemic inflammation that raises BBB permeability. The brain’s circumventricular organs are fenestrated, highly permeable structures. This often enables circulating but non-BBB-crossing mediators to have an impact on brain activity directly. This permeability, however, can also allow for pathogen neuroinvasion during an acute infection. This can happen directly or by an indirect process where host immune cells actively carry intracellular infections into the central nervous system (CNS) [73].

SARS-CoV-2 showed choroid plexus basal (vascular side) epithelial tropism in an organoid model made from human pluripotent stem cells, despite more abundant ACE2 on the apical (CSF) side. Additionally, SARS-CoV-2 led to epithelial destruction and barrier leakage, which may have facilitated neuroinflammation by increasing the entry of immune cells, including those infected in a Trojan horse form, and circulating cytokines into the CNS [73].

Invading the stem cells, pericytes, columnar epithelial cells, and Bowman’s gland cells in the olfactory epithelium using the ACE2 receptor causes persistent filia thinning and volume loss in the olfactory bulb. Although there is limited proof of direct neuroinvasion in these regions, there is a correlation between the geographical distribution of ACE2 receptors and areas of hypometabolism in the brain. Instead, it is thought that these areas suffer from increased levels of oxidative stress, cytotoxic T lymphocyte infiltration, neurodegeneration, and demyelination because of neuroinvasion. These mechanisms probably continue because of persistent viral shedding, particularly in the GI tract, where ACE2 co-regulates dopa-decarboxylase (DDC) and the dopamine metabolic pathway is active [145].

#### 3.8.2. Brain–Gut Axis—Dysbiosis

The intestinal microbiome is a dynamic ecosystem composed of diverse communities of microorganisms, including bacteria, archaea, and fungi. It begins its colonization early in life, undergoing dynamic changes in the first five years and achieving relative stability in adulthood. However, it remains responsive to various factors such as environmental changes, antibiotic use, diet, infections, or diseases. Changes in its composition are associated with multiple health conditions, although causality remains unclear in many cases. The association between microbiome changes and various health conditions like inflammatory bowel disease, obesity, and mental health disorders suggests a potential role of the microbiome in the pathogenesis of these conditions. However, establishing direct causation is challenging due to the complexity of microbiome–host interactions and the numerous influencing factors affecting their composition and functionality.

During the COVID-19 pandemic, research has highlighted the impact of the infection on the intestinal microbiome. Cheng X et al., in a systematic analysis of 16 studies, observed a reduction in intestinal bacterial diversity in patients with acute COVID-19 and those who recovered from the illness [149]. Moreover, a decrease in bacteria producing anti-inflammatory effects mediated by butyrate production (*Megasphaera*, *Dialister*, *Ruminococcus*, *Faecalibacterium*, *Roseburia*, *Lachnospira*, and *Prevotella*) was noted, alongside an increase in microorganisms’ species with pro-inflammatory properties (*Streptococcus, Enterococcus, Corynebacterium, Blautia*, and *Dorea*), leading to increased synthesis of proinflammatory cytokines [149,150].

These disruptions in the gut microbiome persisted into the recovery phase after the resolution of the initial infection, lasting for months, suggesting their potential contribution to PASC. Additionally, an increase in fungi (*Candida albicans*) and inoviruses (*Pseudomonas phages* (Pf1)), as well as a decrease in beneficial bacteria (*Faecalibacterium prausnitzii*, *Eubacterium rectale*, and *Bifidobacterium adolescentis*), have been identified in patients with severe forms of COVID-19, leading to an increased risk of intestinal infections, heightened intestinal permeability, reduced intestinal immune response, and proinflammatory effects [151,152]. Furthermore, a significant increase in proinflammatory pathobionts (*Prevotella species*, *Veillonella species*, *genus Actinomyces*, *Streptococcus anginosus*, and *Gemella sanguinis*) was observed in the oral microbiome of patients experiencing PASC [153]. The changes identified in the oral microbiome of these patients resembled those observed in individuals suffering from chronic fatigue syndrome, suggesting a potential link between the two conditions [153].

In COVID-19, several actions can lead to the onset of dysbiosis, including hypoxia, inflammation, antibiotic use, or decreased tryptophan absorption, as depicted in Figure 4.

The absorption of dietary tryptophan, an essential amino acid absorbed in the small intestine, is mediated by ACE2, the enzyme involved in SARS-CoV-2 cell entry. SARS-CoV-2 leads to a decrease in tryptophan absorption through this pathway [150]. The majority (90%) of tryptophan binds to albumin and crosses the blood–brain barrier, entering the cells of the central nervous system (neurons, astrocytes, and microglia). Some tryptophan remains free in the blood, and the unabsorbed portion serves as a metabolic substrate for intestinal bacteria. Tryptophan is subsequently metabolized via three pathways: the kynurenine pathway, the serotonin pathway, and the Indole pathway [154]. Tryptophan is the sole precursor of serotonin (from which melatonin is later synthesized). Its synthesis primarily occurs in the intestine (enterochromaffin cells) and at neuronal levels (serotonergic neurons in the enteric nervous system and neurons in the central nervous system).

A low level of serotonin can be responsible for brain fog, fatigue, depression, sleep disturbances, headaches, immunosuppression, and constipation. Constipation can, in turn, lead to alterations in the intestinal microbiome. The kynurenine pathway, primarily occurring hepato-renally, is the main metabolic route for tryptophan. It is stimulated by proinflammatory cytokines and corticosteroids, resulting in proinflammatory neurocatabolites (kynurenic acid and quinolinic acid). Almulla A.F. et al., in a meta-analysis of 14 studies, have shown that the kynurenine degradation pathway of tryptophan is highly stimulated in severe COVID-19 [155].

The resulting products are pro-inflammatory, leading to the activation of the aryl hydrocarbon receptor (AhRs), and the direct association with SARS-CoV-2 action on this receptor is responsible for the onset of systemic aryl hydrocarbon receptor activation syndrome (SAAS), responsible for fibrosis, chronic inflammation, impaired cognitive function, increased oxidative stress, thrombosis, and disorders in the functionality of various organs [155]. The third metabolic pathway is the indole pathway, which produces indole derivatives, namely indole-3-propionic acid and indole-3-aldehyde, within the intestinal microbiome. The obtained metabolites have an immune-modulating effect, preventing the harmful effects of free radicals and oxidative stress. Additionally, they have a neuroprotective effect and prevent intestinal dysbiosis [154].

In cases of severe COVID-19, it has been observed that SARS-CoV-2 can continue to be present in the upper respiratory and gastrointestinal epithelium for around 1 month. This prolonged viral shedding is believed to contribute to an imbalance in the brain-gut axis, as the virus tends to deplete symbionts and disrupt the gut microbiome in the gastrointestinal tract. The disturbance of the microbiome is further perpetuated by the continued shedding of the virus. An analysis of human intestinal organoids infected with SARS-CoV-2 has shown that the ACE2 gene is co-regulated with DDC and genes involved in dopaminergic pathway metabolism and the absorption of amino acid precursors for neurotransmitter synthesis. This finding provides additional evidence of an altered brain–gut axis [145].

#### 3.8.3. Long COVID Neuroinflammation

The CNS has seen neuropathological alterations because of COVID-19. The production of pro-inflammatory mediators because of these modifications includes the induction of uncalled-for inflammatory reactions. Inflammation was induced in COVID-19 instances, according to recent clinical findings, and this induction was linked to an elevated level of cytokines, such as IL-1, IL-6, IL-10, and TNF. Previous research has shown that tight junction (TJ) proteins are degraded by cytokines, which change BBB integrity. Emerging data suggest that BBB permeability is increased by TJ degradation, specifically in claudin-5 and zonula occludens (ZO)-1. When the BBB’s integrity is altered, more viruses and cytokines have a chance to cross it and enter the CNS. This causes cerebral immune cells like microglia and astrocytes to become activated, which leads to cytokine-induced neuroinflammation [156,157].

It has been established that astrocytes and microglia have a very important role in neuroinflammation. Microglial cells have been demonstrated to activate in the CNS because of systemic infection, and they are more vulnerable to pathogens than astrocytes. Molecular signals like IL-1 and TNF stimulate astrocytes in response to the activation of microglia. In response to microglial activation, activated astrocytes can release a variety of inflammatory substances, such as TNF, reactive oxygen species (ROS), and nitric oxide (NO). This two-way interaction between astrocytes and microglia intensifies the ‘chain reaction pattern’ of neuroinflammation [156].

TMPRSS2 was also highly expressed by astrocytes in the cerebral cortex of those patients. In general, ACE2 activates the AT1 receptor, which promotes neuroinflammation and oxidative stress. Additionally, it has been demonstrated that the high expression of inflammatory mediators and nuclear factor kappa-light-chain-enhancer (NF-kB) is linked to the elevated level of cathepsin L (CTSL) [156]. It is interesting to note that TMPRSS2 aids virus entry into host cells, according to many studies. All the relevant studies on SARS-CoV-2 neuroinflammation suggest that the virus activated the CNS cells involved in the inflammation [156].

Additionally, COVID-19 encourages mast cell activation, neuroinflammation, and a high intracranial level of proinflammatory cytokines. Alcohol use and drug use disorders are two factors that can exacerbate the neuroinflammatory response caused by SARS-CoV-2, which differs between patients. To give one example, SARS-CoV-2 infection results in TJ loss, mast cell activation, and the production of inflammatory mediators, all of which have the potential to result in neuroinflammation, oedema, and bleeding, particularly in individuals with concurrent neurodegenerative disorders [156]. The impairment of the endothelial barrier and the subsequent increase in blood–brain barrier (BBB) permeability by the SARS-CoV-2 spike protein have been experimentally validated through the utilization of a three-dimensional tissue model of the BBB.

The spike protein of SARS-CoV-2 has the potential to activate brain ECs, leading to an inflammatory reaction that could worsen the breakdown of the blood–brain barrier [156]. The in vitro studies demonstrated that the recombinant SARS-CoV-2 spike protein can decrease the expression of TJ proteins, such as ZO-1, ZO-2, Claudin-5, and junctional adhesion molecule (JAM-2), in human brain microvascular ECs [156]. Changes in junctional protein integrity could affect the BBB as a whole. It is interesting to note that the level of cytokines, including TNF, IL-6, and IL-10, increased when the production of TJ proteins was downregulated. The anti-inflammatory enzyme indoleamine-2, 3-dioxygenase 1 (IDO1) is expressed by a variety of immune cells, including macrophages, monocytes, and microglia. Infection with SARS-CoV-2 leads to abnormal IDO1-mediated inflammation. Therefore, SARS-CoV-2-induced neurological problems may be caused by IDO1 [156].

### 3.9. Myalgic Encephalomyelitis and Chronic Fatigue Syndrome in Long COVID-19

Myalgic encephalomyelitis/chronic fatigue syndrome (ME/CFS) is clinically supported by the presence of a polymorphic, persistent clinical picture (Figure 5) [148,157,158,159,160].

The immunological response to COVID-19 resembles the traditional ME/CFS pattern in key ways. Important aspects of the immunological response to COVID-19 include the loss of basophils and plasmacytoid dendritic cells as well as the depletion of T cells, primarily CD8+ and T cells. The expression of cytokines, particularly IL-6, IL-10, and IFN-induced protein 10 (IP-10, formerly CXCL10), is elevated and closely linked with the course of the disease. IP-10 is of special significance because, like in ME/CFS patients, its concentrations usually remain high throughout the COVID-19 response. The immune system may be facing a bigger strain at higher levels, which could translate into more severe symptoms [158].

Patients diagnosed with COVID-19 exhibited indications of impaired mitochondrial function, alterations in metabolism characterized by heightened glycolysis, and elevated levels of cytokines in peripheral blood mononuclear cells. These observations align with the results of multiple autonomous investigations that have characterized ME/CFS as a state of reduced metabolic activity involving dysfunction in various metabolic pathways. Additionally, there is a possibility that ME/CFS may be classified as a mitochondrial disorder due to heightened mitochondrial harm, diminished adenosine triphosphate (ATP) production, and impaired oxidative phosphorylation. Patients with the COVID-19 infection have compromised mitochondrial activity, making it impossible for them to use this pathway to provide the energy they need. As a result, glycolysis is boosted to make up for the high energy requirements. This is linked to an increase in the inflammatory response. Production of pro-inflammatory cytokines (IL-1 and IL-12) and a propensity for pyroptosis are brought on by inflammation activation. After pyroptosis, cell-free mitochondrial DNA is released, which exacerbates both local and systemic inflammation [158].

Figure 6 presents the clinical picture of Long COVID.

## 4. Conclusions

Long COVID is a complicated and multidimensional illness that affects a large proportion of those recovering from an acute COVID-19 infection. It has been linked to a variety of symptoms and problems, including chronic fatigue, cognitive impairment, respiratory troubles, cardiovascular irregularities, and psychological discomfort, according to Figure 6. The precise processes behind Long COVID are yet unknown. Pathology presents considerable issues for healthcare practitioners since it necessitates a multidisciplinary approach to diagnosis, treatment, and rehabilitation. Future research should concentrate on elucidating the pathogenesis of Long COVID, including the function of viral reservoirs, immune response dynamics, and potential long-term organ damage. To produce evidence-based guidelines for the diagnosis, treatment, and management of Long COVID, clinical trials are required. The research should also emphasize the development of biomarkers that can help in the condition’s early detection, prognosis, and monitoring. This COVID-19 complication should be the focus of public health campaigns, with a focus on increasing awareness, advocating preventative measures, and offering support and services to patients with long-term symptoms. Collaboration between researchers, healthcare professionals, and patient groups is critical for effectively addressing the issues posed by Long COVID and improving patient outcomes.

## Figures and Tables

**Figure 1 diagnostics-14-00244-f001:**
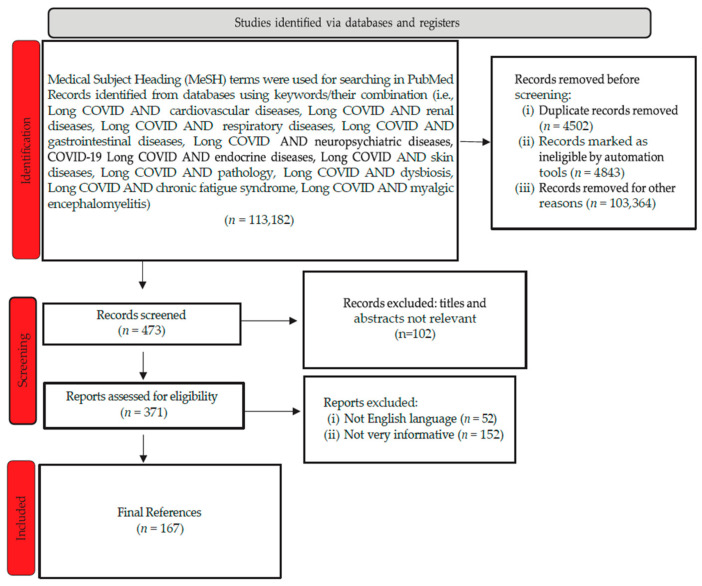
The PRISMA flowchart for the selected studies.

**Figure 2 diagnostics-14-00244-f002:**
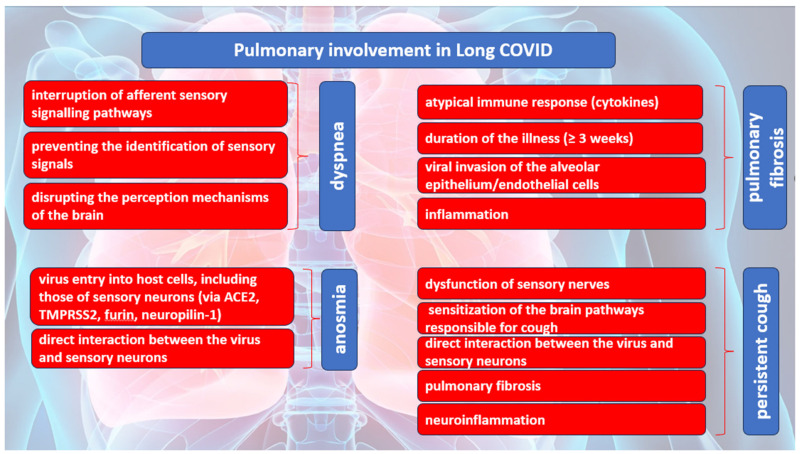
The causes of the most common pulmonary manifestations in Long COVID.

**Figure 3 diagnostics-14-00244-f003:**
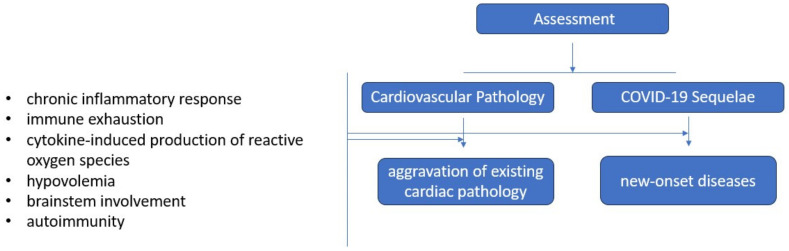
Possible mechanisms of cardiac damage in COVID-19.

**Figure 4 diagnostics-14-00244-f004:**
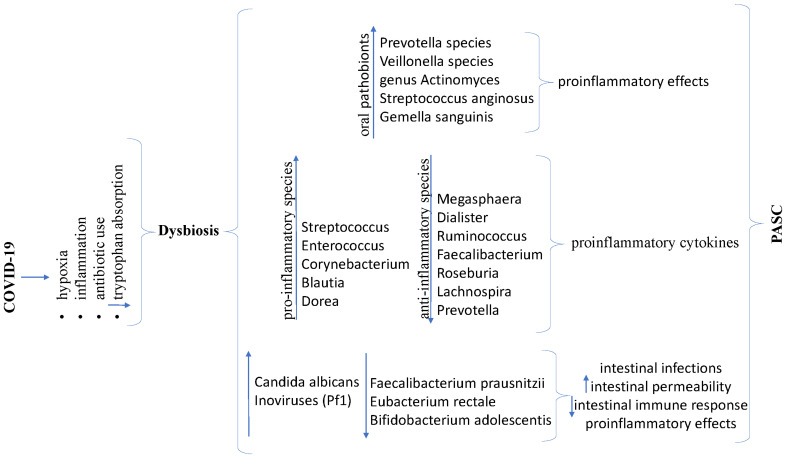
Dysbiosis in COVID-19. PASC—Post-Acute Sequelae of COVID-19, Pf1—Pseudomonas phages, ↓—decrease, ↑—increase.

**Figure 5 diagnostics-14-00244-f005:**
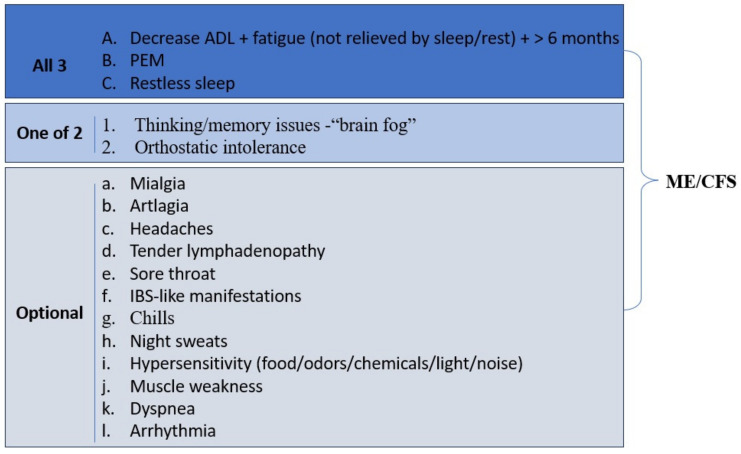
ME/CFS clinical picture. ADL—activity of daily living, PEM—post-exertional malaise, IBS—irritable bowel syndrome, and ME/CFS—Myalgic encephalomyelitis/chronic fatigue syndrome.

**Figure 6 diagnostics-14-00244-f006:**
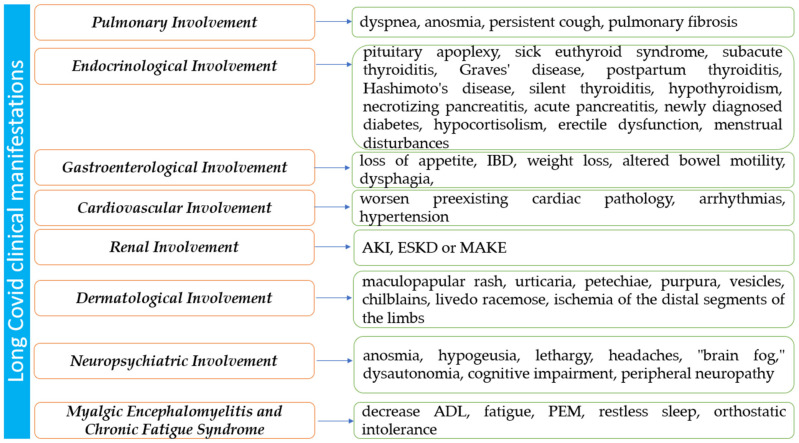
Clinical picture of Long COVID. IBD—inflammatory bowel disease, AKI—acute kidney injury, ESKD—end-stage kidney disease, MAKE—major adverse kidney events, ADL—activity of daily living, and PEM—post-exertional malaise.

**Table 1 diagnostics-14-00244-t001:** Long-term effects on the endocrine system.

Gland	Clinical Manifestation
Pituitary gland	Pituitary apoplexy
Thyroid gland	Euthyroid sick syndrome Subacute thyroiditis Graves’ disease Postpartum thyroiditis Hashimoto’s disease Silent thyroiditis
Pancreas	Acute pancreatitis Necrotizing pancreatitis Impaired glucose tolerance Insulin resistance Diabetes mellitus Severe metabolic complications (diabetic ketoacidosis) Hyperglycemia Abnormalities level of amylase/lipase
Adrenal glands	Necrosis Hypocortisolism Cross-reactive antibodies against endogenous adrenocorticotropic hormone
Gonads	Hypogonadism Erectile dysfunction Menstrual disturbances

## Data Availability

Not applicable.

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
