# Peer review of "The Multisystem Impact of Long COVID: A Comprehensive Review"

_diagnostics, 2024, doi:10.3390/diagnostics14030244_

Round 1
Reviewer 1 Report
Comments and Suggestions for Authors
The definition used for long COVID is very important for the current research. The following statement needs an abbreviation "The description given for Long COVID by the World 48 Health Organization (WHO) ..."
Also, the terms used for the literature search is very important. How were the following terms chosen? ""cardiovascular diseases or renal diseases or respiratory diseases or gastrointestinal diseases or neuro-psychiatric diseases or endocrine diseases or skin diseases or pathology or dysbiosis or chronic fatigue syndrome or myalgic encephalomyelitis."
The guidelines for systematic reviews demand that the search should be done in at least 2 databases, it looks like only one was used: the name Pubmed via Medline should be noted in Materials and Methods (search in Scopus or Web of Science is advisable).
It's not clear how was the information from the 167 articles searched. What kind of data were extracted.
The conclusions seem too general for a synthesis of literature. Maybe more precise ideas could be summarized. Also, it could be useful a Table with all possible long COVID clinical manifestations could be summarized (with reference for each of the symptoms listed).
Author Response
|
Response to Reviewer 1 Comments |
|
Thank you very much for taking the time to review this manuscript. Please find the detailed responses below and the corresponding revisions highlighted (yellow) in the re-submitted revised manuscript. Also, small corrections have been made in English, and they are marked in yellow background. |
|
Comments 1: The definition used for long COVID is very important for the current research. The following statement needs an abbreviation "The description given for Long COVID by the World 48 Health Organization (WHO) ..." |
|
Response 1: Thank you for the observations. I made the necessary changes. “Long COVID, according to the World Health Organization (WHO), represents any clinical manifestation, present in the first three months after acute COVID-19, minimally evolving over two months, without another identifiable cause [1].” Please see L47 of the main manuscript. |
|
Comments 2: Also, the terms used for the literature search is very important. How were the following terms chosen? ""cardiovascular diseases or renal diseases or respiratory diseases or gastrointestinal diseases or neuro-psychiatric diseases or endocrine diseases or skin diseases or pathology or dysbiosis or chronic fatigue syndrome or myalgic encephalomyelitis." |
|
Response 2: The selection of terms used in the literature search was carried out carefully and in line with our research objectives on long COVID syndrome. The terms were chosen to encompass a broad range of possible manifestations associated with this syndrome. |
|
Comments 3: The guidelines for systematic reviews demand that the search should be done in at least 2 databases, it looks like only one was used: the name Pubmed via Medline should be noted in Materials and Methods (search in Scopus or Web of Science is advisable). Response 3: Thank you for your observation. The research was conducted on several databases. Scopus and Web of Science databases were omitted by mistake during writing. Please see L68-69 of the main manuscript. Comments 4: It's not clear how was the information from the 167 articles searched. What kind of data were extracted. Response 4: Related information about various clinical manifestations reported on various systems was extracted, looking for data about the mechanisms underlying these manifestations. Comments 5: The conclusions seem too general for a synthesis of literature. Maybe more precise ideas could be summarized. Also, it could be useful a Table with all possible long COVID clinical manifestations could be summarized (with reference for each of the symptoms listed). Response 5: Thank you for the observation. At your request and that of the second reviewer, we have added a graph with the requested clinical data (please see newly added Figure 6). |
Reviewer 2 Report
Comments and Suggestions for Authors
The authors have provided a comprehensive review paper based on published data and literature on impact of long covid 19 on diferent organ systems. The data extraction and presentation of the results are good. Although there are many other review papers on the same topic, but I see this review paper covering some new angels on long covid. I recommed this paper for publishing. Adding a schematic graphical image showing the basic impacts of covid in each organ can improve the paper as graphical abstract for the findings in the literature
Comments on the Quality of English LanguageThe English language of the paper is in a good shape.
Author Response
Thank you for the observation. We have added the requested graphic (please see Figure 6 in the main manuscript) as per your suggestion. I hope this addition enhances the quality and accessibility of the article.
Reviewer 3 Report
Comments and Suggestions for Authors
This paper provides an exhaustive review covering various aspects of COVID-19 infections.
It appears that the paper is now prepared for publishing.
Author Response
Thank you very much for taking the time to review this manuscript. Please find the detailed responses below and the corresponding revisions highlighted (yellow) in the re-submitted revised manuscript. Also, small corrections have been made in English, and they are marked in yellow background.
Thank you very much for taking the time to review this manuscript.